# Identification of Secreted Protein Gene-Based SNP Markers Associated with Virulence Phenotypes of *Puccinia striiformis* f. sp. *tritici*, the Wheat Stripe Rust Pathogen

**DOI:** 10.3390/ijms23084114

**Published:** 2022-04-08

**Authors:** Qing Bai, Meinan Wang, Chongjing Xia, Deven R. See, Xianming Chen

**Affiliations:** 1Department of Plant Pathology, Washington State University, Pullman, WA 99164-6430, USA; qing.bai@wsu.edu (Q.B.); meinan_wang@wsu.edu (M.W.); chongjing.xia@wsu.edu (C.X.); deven.see@usda.gov (D.R.S.); 2Wheat Research Institute, School of Life Sciences and Engineering, Southwest University of Science and Technology, Mianyang 621010, China; 3U.S. Department of Agriculture, Agricultural Research Service, Wheat Health, Genetics, and Quality Research Unit, Pullman, WA 99164-6430, USA

**Keywords:** correlation coefficient, *Puccinia striiformis* f. sp. *tritici*, secreted protein gene, SNP markers, wheat stripe rust, virulence

## Abstract

Stripe rust caused by *Puccinia striiformis* f. sp. *tritici* (*Pst*) is a destructive disease that occurs throughout the major wheat-growing regions of the world. This pathogen is highly variable due to the capacity of virulent races to undergo rapid changes in order to circumvent resistance in wheat cultivars and genotypes and to adapt to different environments. Intensive efforts have been made to study the genetics of wheat resistance to this disease; however, no known avirulence genes have been molecularly identified in *Pst* so far. To identify molecular markers for avirulence genes, a *Pst* panel of 157 selected isolates representing 126 races with diverse virulence spectra was genotyped using 209 secreted protein gene-based single nucleotide polymorphism (SP-SNP) markers via association analysis. Nineteen SP-SNP markers were identified for significant associations with 12 avirulence genes: *AvYr1*, *AvYr6*, *AvYr7*, *AvYr9*, *AvYr10*, *AvYr24*, *AvYr*27, *AvYr32*, *AvYr43*, *AvYr44*, *AvYrSP*, and *AvYr76*. Some SP-SNPs were associated with two or more avirulence genes. These results further confirmed that association analysis in combination with SP-SNP markers is a powerful tool for identifying markers for avirulence genes. This study provides genomic resources for further studies on the cloning of avirulence genes, understanding the mechanisms of host–pathogen interactions, and developing functional markers for tagging specific virulence genes and race groups.

## 1. Introduction

Stripe rust (yellow rust), caused by *Puccinia striiformis* Westend. f. sp. *tritici* Erikss. (*Pst*), is a destructive disease that occurs throughout the major wheat-growing regions of the world [1,2,3,4,5]. This obligate biotrophic fungal pathogen is highly variable due to the capacity of virulent races to undergo rapid changes in order to circumvent resistance in wheat cultivars and genotypes and to adapt to different environments [1,3,6,7,8,9,10,11,12]. In our group, more than 320 virulent races have been identified with *Pst* collections from the U.S. since the 1960s and ten other countries from 2007 to 2020 [13,14,15,16]. Furthermore, some new molecular groups specific to one or more countries have been identified through the analysis of population structure and differentiation, and these currently small groups have the potential to threaten wheat production in other countries [17]. Therefore, it is important to study virulence and genotype changes and the molecular mechanisms behind the rapid changes.

To investigate *Pst* race evolution, researchers have used a combination of the classical theory and modern sequencing technology. Since the gene-for-gene recognition between host resistance and pathogen avirulence genes was phenotypically demonstrated by Flor in the middle 20th century [18], research on identification and function demonstration of avirulence (*Avr*) genes has been conducted in rust pathogens [19,20,21]. In the flax rust fungus *Melampsora lini*, the avirulence gene *AvrL*567 was first molecularly characterized through genetic mapping along with a cosegregating cDNA probe [20,21]. In recent years, a combination of different technologies, including genetic mapping and genomic approaches, has become popular for studying *Avr* genes in rust fungi. Using this approach, high-density genetic maps were constructed for poplar rust fungus (*Melampsora larici-populina*) [22] and *Pst* [23,24]. In Xia et al. (2020), the QTL analysis of a sexual population mapped six *Avr* genes in three linkage groups and identified a genomic cluster at a single contig containing four *Avr* genes (*AvYr*7, *AvYr43*, *AvYr44*, and *AvYrEpx2*) [23].

Correlation analysis between genetic variations and virulence/avirulence phenotypes has been used to study host–pathogen interactions [25,26]. In rust fungi, highly expressed secreted protein (SP) genes have been demonstrated to include some proteins with important pathogenicity functions [19]. The SP gene-derived SNP markers (SP-SNP) have been used to study population structures and to tag specific virulence genes. In our group, Xia et al. (2016) attempted to identify *Avr* candidate genes in *Pst* and indicated that association analysis of genetic variations (SP-SNPs) and virulence/avirulence phenotypes can be used to identify markers for *Avr* genes. Xia et al. (2017) further took advantage of comparative genomics and correlation analysis and identified more than 900 *Pst*-specific SP genes and 73 *Avr* candidate genes. In a most recent study, 62 additional avirulence candidate genes significantly associated with 16 avirulence genes were identified by means of a comparison of the genomic variations in 30 mutant isolates derived from ethyl methanesulfonate (EMS) mutagenesis with respect to their progenitor isolate [27,28]. In addition, some virulence factors have also been reported in *Pst*, such as PstSCR1 activating immunity in non-host plants [29], Pst_8713 involved in enhancing *Pst* virulence [30], and Pst18363 as an important pathogenicity factor in *Pst* [31]. However, no known *Avr* genes have been identified in *Pst* so far.

In our group, we have utilized SNPs from sequence data to distinguish different *P. striiformis* isolates [32], develop SNP markers for SP genes [33], and use them for characterizing populations, constructing linkage maps, studying virulence, and determining mechanisms for variations [24,27,33,34,35]. Xia et al. (2017) identified more than 900 *Pst*-specific SP genes and we designed hundreds of primers based on the SP genes. Therefore, the objectives of the present study were to (1) develop more SNP markers using genomic sequencies of *Pst* SP genes, (2) further characterize the U.S. and international *Pst* isolates using the new SP-SNP markers, and (3) identify SP-SNPs associated with avirulence genes.

## 2. Results

### 2.1. Distribution of Avirulence/Virulence Phenotypes

The infection type (IT) data of the selected 157 *Pst* isolates are provided in Appendix A and the distribution of avirulence and virulence phenotypes determined for the 18 *Yr* single-gene lines are shown in Figure 1. Of the 18 *Yr* gene lines, 2 (*Yr5* and *Yr15*) had only the avirulence phenotype, and thus the identification of markers for virulence in these two resistance genes was not possible. Therefore, these genes were excluded from further analyses. The remaining 16 *Yr* genes had the less frequent phenotype, all above the 0.05 frequency value, and therefore they were suitable for association analysis.

### 2.2. SP-SNP Markers

After eliminating SP-SNP markers with a minor allele frequency (MAF) <5% and a missing rate >50%, 209 SP-SNPs were retained for subsequent analyses (Appendix A). The MAF of the 209 SP-SNPs based on the 157 isolates ranged from 0.05 to 0.50 with a mean of 0.33. The results indicated that the 209 markers were suitable for genotyping *Pst* isolates.

### 2.3. Population Structure

Principal component analysis (PCA) conducted using the GAPIT program indicated that, using the 209 SP-SNP markers, the 157 *Pst* isolates were optimally separated into three groups (Figure 2A), and PC1 and PC2 explained 19.23% and 7.65%, respectively (Figure 2B). The first two PCs separated the isolates into three molecular groups (MGs). The detailed relationships among the isolates with the country of origin for each of the three MGs are shown in Figure 3. The first group (MG1) was the most diverse, containing the isolates from all countries except Canada. The second group (MG2) was the smallest and was closely related to MG1, containing isolates mainly from Ecuador (80%). Mostly distinct from MG1 and MG2, the third group (MG3) contained isolates from China and countries in North America (the U.S., Canada, and Mexico) and South America (Ecuador). The Efficient Mixed Model Association (EMMA) algorithm was used to establish a kinship matrix and a heat map of values in the kinship matrix showed three groups (Appendix A). These different analyses consistently revealed three molecular groups, and the structures were considered in the following association analysis.

To compare the clusters of the 157 *Pst* isolates in the present study with our previous studies, in which more isolates were genotyped using 14 SSR markers [17,36], another phylogenetic tree with the same 157 *Pst* isolates based on the 14 SSR markers was generated (Appendix A). The clusters in the phylogenetic tree based on the SSR markers (Appendix A) had both similarities and differences with the phylogenetic tree based on the 209 SP-SNP markers (Figure 3). There were three main molecular groups based on both SSR and SP-SNP markers, while the differences were the clustering of the isolates in each group. Most of the isolates in the first group (MG1) based on the SSR markers were also clustered in the MG1 based on the 209 SP-SNP markers. The second group (MG2) based on the SSR markers contained two subgroups: one subgroup (indicated by the red arrow) contained the isolates that were clustered in the largest group (MG1) based on the 209 SP-SNP markers; the second subgroup (indicated by the black arrow) contained the isolates mainly from Ecuador (9 out of 13 isolates), which was similar to the second group (MG2) based on the 209 SP-SNP markers. The third group (MG3) based on the SSR markers was most distinct from MG1 and MG2 and was similar to the MG3 in the phylogenetic tree based on the 209 SP-SNP markers. Another similarity between the two phylogenetic trees with respect to MG3 was that they both contained most of the Chinese isolates, including 11 of the 12 isolates based on the SSR markers and 8 of the 12 isolates based on the SP-SNP markers. The correlation coefficient of the distance similarity matrices constructed by the two sets of markers was 0.878 (*p* < 0.01). Overall, the high similarity of the clusters based on the SP-SNP and the SSR markers confirmed the population structure and the effectiveness of the SP-SNP markers used in the population genetic analysis.

### 2.4. SP-SNPs Significantly Associated with Avirulent Genes

The analysis of a mixed linear model with PCA and kinship identified 19 SP-SNP markers significantly associated with 12 avirulence genes (corresponding to 12 *Yr* genes) as having *p-*values < 0.01 (Table 1). One marker each was found for *AvYr7*, *AvYr10*, *AvYr32*, *AvYr**76*, and *AvYr**SP*; two markers each for *AvYr6*, *AvYr24*, and *AvYr**43*; three markers for *AvYr**44*; and four markers each for *AvYr1*, *AvYr9*, and *AvYr27*. The detailed information on the supercontig and position, *p*-value for the association, MAF, percentage of variation explained (PVE), and alleles of nucleotides for each of the markers is presented in Table 1. Of the 19 SP genes, 8 were predicted to be effectors (Appendix A). The QQ and Manhattan plots for each of the marker–avirulence gene associations are shown in Figure 4. The QQ plot shows the deviation of the observed *p*-values from the null hypothesis that the SP-SNP is not associated with the avirulence gene. Markers along the diagonal line were not associated, while those away from the diagonal line were associated with the avirulence gene. The Manhattan plot shows the significant *p*-values above the threshold (*p* < 0.01, −log_10_(*p*) > 2) associated with the avirulence gene. Some of the SP-SNPs were significantly associated with two or more avirulence genes; for example, marker SP.SNP.SC.120.10252 was associated with *AvYr10*, *AvYr24*, and *AvYr32* (Table 1, Figure 4). The 19 markers associated with 12 avirulence genes resulted in 27 significant marker–avirulence gene associations. The association of a single marker with multiple avirulence genes was likely due to the correlation of the phenotypic data of the avirulence genes. To test the hypothesis, correlation coefficients of the virulence phenotypes for different avirulence genes were estimated, and the results are presented in Figure 5. The correlation coefficients between most pairs of the avirulence genes were relatively low or moderate. However, high correlations were observed between *AvYr10* and *AvYr32* (r = 0.88, *p* < 0.001), between *AvYr24* and *AvYr32* (r = 0.81, *p* < 0.001), and between *AvYr10* and *AvYr**24* (r = 0.77, *p* < 0.001). The PVE values of the 19 SP-SNP markers were low to moderate, ranging from 0.06 to 0.21, but all were significant (*p* < 0.01) (Table 1).

### 2.5. Accuracy and Sensitivity for Detecting Avirulence/Virulence Genes

The accuracies for the detections of avirulence/virulence ranged from 50.39% in the test of the marker SP.SNP.SC.187.104441 with the avirulence gene *AvYr27* to 94.90% in the test of the marker SP.SNP.SC.120.10252 with the avirulence gene *AvYr32* (Appendix A). The higher accuracy indicated that the SP-SNP markers could be used to differentiate more accurately between virulent alleles and avirulent alleles. The sensitivities of all the tests were all higher than 90% except for that for the marker SP.SNP.SC.241.57435 (78%) with the avirulence gene *AvYr9*. A marker with a higher sensitivity should provide a higher rate of correct predictions of virulent or avirulent phenotypes.

## 3. Discussion

The results of the present study confirmed that association analysis can be a powerful tool for identifying molecular markers associated with *Pst* avirulence genes, especially when the markers are developed from polymorphic SP-SNPs, as first reported by Xia et al. (2016). Different from the previous study that utilized non-selected isolates from two years and only from the U.S., the present study used 157 isolates selected from nine countries over an eight-year period. These isolates were identified as 126 races using 18 *Yr* single-gene lines, thus representing diverse avirulence/virulence profiles. The 157 isolates were also previously identified as 157 multi-locus genotypes (MLGs) using 14 simple sequence repeat (SSR) markers, representing major molecular groups [17,36]. The highly diverse isolates should be more suitable for association analysis for identifying markers associated with avirulence genes. From the 209 SP-SNP markers used to genotype the 157 isolates, we found 27 significant (*p* < 0.01) marker–trait associations involving 12 *AvYr* genes and 19 SP-SNP loci (Table 1). The number of avirulence genes with associated markers is relatively high compared with previous association analysis studies [23,35,37] and this can be attributed to the large number of SP-SNPs and the selected isolates that have relatively balanced virulence/avirulence ratios. Similar to previous studies, we also found that some markers were associated with different avirulence genes, suggesting that these avirulence genes are located in a gene cluster [23,24,35]. For example, the SP-SNP marker SP.SNP.SC.120.10252 was significantly associated with *AvYr10*, *AvYr24*, and *AvYr32*, which is supported by the phenotypic correlations between these avirulence genes [38,39].

In the present study, we identified SP-SNP markers for 12 of the 18 avirulence genes corresponding to the 18 *Yr* single genes in the differentials but did not find markers for the remaining 6 genes. As none of the *Pst* isolates were virulent to either *Yr5* or *Yr15*, it was not possible to identify markers for their corresponding avirulence genes. However, it was possible to find markers for *AvYr8*, *AvYr17*, *AvYrTr1*, and *AvYrExp2*, as the 157 isolates showed relatively balanced virulence/avirulence ratios (Figure 1). Failing to identify markers for these genes may be due to the limitation of the 209 SP-SNPs used in the present study, which do not cover the entire genome. Secondly, only SP-SNPs showing co-dominant polymorphisms among the 14 whole-genome sequenced isolates used in the previous study [37] were used to design primers for SP-SNP markers. Avirulence/virulence alleles that have presence/absence or indel polymorphisms likely escaped from the test with these markers.

Among the 12 avirulence genes with associated SP-SNP markers, 7 had two or more markers in different supercontigs. The supercontigs were assigned based on the PST-78 reference genome, the best available at that time [37]. Some of the supercontigs may be linked but some of them may be far away. Further annotation of these SP-SNPs is needed to determine their genomic and molecular relationships. The genomic relationships of the markers associated with the same avirulence gene could be improved by annotation using the high-quality genomes recently established [40,41,42]. Nevertheless, different supercontig markers for a single avirulence gene may indicate that the avirulence phenotype is controlled by different genes in different genomic regions. Two-gene controlled virulence was demonstrated by genetic analysis of sexually produced *Pst* populations [23,24].

In an association study, individuals should be distinct [43]. However, as a predominantly asexually reproduced fungal pathogen, population structures of *Pst* have been reported in previous studies [17,36,44,45]. In a structured population, individuals with differences in allele frequencies between sub-populations due to ancestries which are unrelated to the trait of interest can cause false-positive associations in association studies [43]. To address this problem, PC analysis, which can effectively determine population structure, was used in the present study prior to the association analysis. However, PC analysis only accounts for fixed effects of genetic ancestry and does not account for relatedness between individuals. Therefore, the mixed-model approach, which utilizes both fixed effects (candidate SNPs and fixed covariates) and random effects (the genotypic covariance matrix) involving kinship and cryptic relatedness, was used in the further association analysis. In addition, false-positive associations might also be caused by statistical fluctuations governed by chance in multiple-hypothesis testing [46]. To control statistical fluctuations, various statistical approaches have been proposed, including Bonferroni correction and estimation of the false detection rate (FDR), which are two common correction methods [47,48]. In the present study, Bonferroni correction was applied for the suggestive threshold *p*-value (*p* = 1/*Ne*), where *Ne* represents the effective number of SNP markers [49]; therefore, *p* = 1/*Ne* = 1/209 = 0.0047≈0.01 was chosen (-log_10_ (0.01) = 2) as the threshold for significance in the association analysis. Besides these two approaches, replicating genotype–phenotype associations in larger and independent populations is another way of establishing the credibility of genome-wide association studies (GWAS) [50]. Therefore, in future studies, we will genotype a large number of *Pst* isolates with the SP-SNPs identified in the present study to confirm the associations and also genotype the isolates using additional SP-SNPs to identify more markers.

By using the SP-SNP markers, the *Pst* isolates in the present study were clustered into molecular groups similar to those in our previous studies, in which more isolates were genotyped using the set of 14 SSR markers [17,36]. For example, isolates EC15-039, EC15-019, EC15-022, EC15-26, EC15-27, EC15-30, and EC16-019, which were from Ecuador, were separated into a molecular group different from the groups of other isolates in the present study, consistent with the results of our previous study. However, the SP-SNP markers significantly associated with avirulence genes can provide information for virulence evolution and may be used to tag virulence genes [33,35,37]. Every year, we establish more than 300 isolates from stripe rust collections throughout the U.S. and phenotype them for virulence/avirulence profiles [14,15,39]. Genotyping the isolates with the SP-SNPs may confirm the avirulence-associated SP-SNPs identified in the present study and previous studies [33,35,37], which may lead to the identification and cloning of the avirulence genes.

All SP-SNP markers used in the present study were developed based on the SP genes characterized by Xia et al. (2017) through analyzing the whole-genome sequences of 14 *Pst* isolates [37]. All SP genes were annotated for identification of polymorphic SNPs and predicted for effectors using the EffectorP program. In the present study, among the 19 SP-SNP markers significantly associated with *Avr* genes, 8 were predicted as *Pst* effectors with high confidence (Table 1), indicating their possible involvement in interactions with host plants [37]. However, most of the SP genes were predicted to encode hypothetical proteins (Appendix A), while only a few were annotated as being involved in different biological processes. None of them is known to be involved in pathogenicity. One SP gene, PSTG_15874, which was significantly correlated with avirulence to *Yr9*, has a high homology with a phosphatidylinositol/phosphatidylglycerol transfer protein (PG/PI-TP) gene. The PG/PI-TP proteins belong to the ML (MD-2-related lipid-recognition) domain family and have been shown to bind phosphatidylglycerol and phosphatidylinositol, but the biological significance of these proteins is still not clear [51].

For association analysis, high resolution mapping depends on the number of markers as well as on linkage disequilibrium (LD) decay [46,48,52]. LD is the nonrandom association of alleles at different loci that plays a central role in association analysis [48]. Therefore, for establishing the credibility of the association analysis, the LD decay of the *Pst* genome also should be estimated [35]. Xia et al. (2020) has developed a high-quality map for *Pst* comprising 41 lineage groups, which will enable us to identify *Avr* candidates in narrow genome regions and study their functions [31]. In the present study, even though we identified more SP-SNP markers than the previous study [35], we were still unable to estimate LD decay because of unknown physical distances between them. Moreover, due to budgetary and time limitations, we only tried the primers of the first 390 of the more than 900 SP genes and used 209 successful markers in the present study. For future studies, we will use the remaining SP genes. Alternately, we can genome-sequence and RNA-sequence the isolates used in the present study to identify avirulence genes involved in interactions with the *Yr* genes.

The accuracy and sensitivity of these SP-SNP markers for detecting the associated avirulence/virulence genes should provide more information for the subsequent use of these markers in monitoring individual virulence factors and race changes in the pathogen population. The higher the accuracy and sensitivity of the markers, the more useful the markers are for predicting the virulence phenotype. For example, high accuracies were found in the tests of the same marker SP.SNP.SC.120.10252 associated with the avirulence genes *AvYr10* (92.86%), *AvYr24* (92.86%), and *AvYr32* (94.90%), as the majority of isolates are avirulent to the corresponding *Yr* genes. However, caution should be taken when using the markers. Further testing of the markers for the opposite alleles for the virulence phenotypes needs more isolates virulent to these *Yr* genes, which may take years to accumulate as virulence frequencies for these genes have been low in the past several decades [13,14,15,16,39].

The present study aimed to identify additional SP-SNP markers associated with avirulence genes from the sequences of SP genes. Virulence-associated markers can be used to monitor virulence and race changes in the pathogen population. However, the Ion Protein sequencing approach used in the present study for identifying SNPs is not efficient for routine tests for monitoring virulence changes. To overcome this drawback, SNP markers can be converted to Kompetitive Allele Specific PCR (KASP) markers, which should be used for efficiently monitoring *Pst* populations. The SP-SNP markers identified in the present study, together with the other nearly 100 SP-SNP markers associated with *Pst* virulence in previous studies [27,31,35,37,44], will be converted to KASP markers for further confirmation of the association of the SP-SNP markers with their avirulence genes and also for establishing a set of virulence-related markers for monitoring changes in both virulence and molecular genotypes in the *Pst* population. Compared to SNP genotyping, KASP markers are relatively cheap and easy to use, as demonstrated in our program for stripe rust resistance genes in wheat [53,54,55,56,57]. The present study provides genomic resources for further marker development and research on host–pathogen interactions, as well as pathogen population dynamics.

## 4. Materials and Methods

### 4.1. Isolate Selection

A total of 157 *Pst* isolates used in the present study were selected based on races and MLGs from *Pst* collections from the U.S. and eight other countries assembled in the period 2010-2020 [13,17,36]. These isolates had different MLGs and represented 126 races (Table 2).

### 4.2. Virulence Data

The *Pst* isolates were tested for their avirulence/virulence patterns using the 18 *Yr* single-gene lines, and the virulence data have been reported previously [13,14,15,39]. The avirulence or virulence of isolates on a particular *Yr* resistance gene line was represented by an infection type (IT), which was scored using a 0-to-9 scale, with 0 as the most avirulent and 9 as the most virulent [58]. For the association analysis in the present study, the phenotype of an isolate for a particular *Yr* gene was defined as avirulent when IT was 0–6 and virulent when IT was 7–9 [14]. To reduce phenotypic variation within avirulent and virulent classes, isolates with ITs 0–2 for avirulent reactions and with ITs 7–9 for virulent reactions were selected, and thus intermediate ITs (3–6) were mostly avoided (Figure 1, Appendix A). The 18 resistance genes were *Yr1*, *Yr5*, *Yr6*, *Yr7*, *Yr8*, *Yr9*, *Yr10*, *Yr15*, *Yr17*, *Yr24*, *Yr27*, *Yr32*, *Yr43*, *Yr44*, *YrExp*2, *YrSP*, *YrTr1*, and *Yr76*. Their corresponding avirulence genes were symbolized as *AvYr1*, *AvYr5*, and so on. Generally, the selected 157 isolates represented a relatively balanced virulent–avirulent profile for the majority of the *Yr* genes.

### 4.3. DNA Extraction

DNA extraction from urediniospores of the *Pst* isolates was described in our previous study [36]. The concentration of the DNA stock solution was determined using a ND-1000 spectrophotometer (Bio-Rad, Hercules, CA, USA), and the quality was checked in a 0.8% agarose gel. A work solution of 0.5 ng µL^−1^ was made from the stock solution by adding sterile deionized water for use as a DNA template in polymerase chain reaction (PCR).

### 4.4. Development of SP-SNP Markers

The genomic sequences of *Pst*-specific SP genes containing SNPs among 14 whole-genome sequenced *Pst* isolates [37] identified using the IGV software (https://igv.org/app, accessed on 27 September 2020) were used to develop SP-SNP markers. The SP-SNP primers were designed using the Sequenom MassArray Assay Design 4.0 software (Sequenom, San Diego, CA, USA). The primers were modified by adding barcodes and specific sequences compatible with the Ion Torrent Proton System (LifeTechnologies, Carlsbad, CA, USA). The locus-specific forward primers for the first round of PCR were tailed with an M13-derived sequence (GATGTAAAACGACGGCCAGTG) at the 5′-end to enable the addition of barcoded adapters during the second round of PCR. The Ion truncated P1 adapter sequence (CCTCTCTATGGGCAGTCGGTGAT) was concatenated to the 5′-end of the locus-specific reverse primers (Appendix A). For the second round of PCR, the forward fusion primer consisted of, from 5′ to 3′, the standard Ion A adapter sequence (CCATCTCATCCCTGCGTGTCTCCGACTCAG), a unique barcode with 10–12 nucleotides, followed by the M13 tail sequence (Appendix A). A combination of different barcodes with the M13 tail proved the flexibility required to multiplex the same set of markers in different samples. The reverse primer for the second round of PCR was the Ion truncated P1 adapter sequence.

### 4.5. Isolate Genotyping

Prior to sequencing, library construction, sample purification, size selection, and quantification were conducted using the standard procedures [59,60,61]. The library construction included two steps of PCR. In the first step of PCR, each reaction (10 µL) contained 1.0 µL of McLab 10× Taq PCR buffer (McLab, San Francisco, CA, USA), 0.45 µL of 25 mM MgCl_2_, 1 µL 5 mM dNTP, 1 µL of 125 nM primer pool, 0.2 µL 5 U/µL McLab HoTaq polymerase (McLab), 2 µL of 0.5 ng/µL DNA (total 1 ng), and 4.35 µL sterile ddH_2_O. The amplification cycles and conditions were 94 °C for 1 min for initial denaturation; 35 cycles of 94 °C for 20 s, 56 °C for 2 min, and 68 °C for 30 s; and 3 min of final extension at 72 °C. The first-step PCR products were diluted at 1:1 with ddH_2_O into a 96-well plate for the second step of PCR. In the second step of PCR, each reaction (6 µL) contained 0.5 µL of McLab 10× Taq PCR buffer, 0.2 µL of 25 mM MgCl_2_, 0.025 µL 100 mM dNTP, 0.2 µL P1 reverse primer, 0.2 µL 5 U/µL McLab HoTaq polymerase, 2 µL of diluted PCR products from the first-step PCR, 0.875 µL sterile ddH_2_O, and 2 µL of 2 µM barcoded adapters 289–384 and 385–480. The amplification cycles and conditions were 94 °C for 1 min for initial denaturation; 15 cycles of 94 °C for 15 s, 60 °C for 30 s, and 72 °C for 1 min; and 3 min of final extension at 72 °C.

The second-step PCR products were cleaned using a QIAquick PCR Purification kit (Qiagen, Hilden, Germany). The size selection of the libraries was first performed on a 4% E-Gel SizeSelect Gel (Life Technologies, Carlsbad, CA, USA) and then on a 2% E-Gel1 SizeSelect™ Gel (LifeTechnologies) to select PCR fragments from 140 bp to 250 bp. Purified libraries were quantified using a Qubit1 dsDNA HS assay kit (LifeTechnologies), diluted to the appropriate concentration as recommended by LifeTechnologies (minimum 80 pmol/L) and a size distribution between 185 bp to 260 bp. The diluted sample libraries were prepared for sequencing using an Ion Torrent Proton NGS platform (Life Technologies).

### 4.6. Data Analyses

The IT distribution and density of isolates on the 18 *Yr* single-gene differentials were determined using the *vioplot* package in the R program 4.1.1. The genotypic data from an initial set of 390 SP-SNP markers were subjected for quality control by excluding the markers with >50% missing data and a minor allele frequency (MAF) less than 0.05. The filtered set of 209 SP-SNP markers was finally retained, and the remaining missing data were imputed using the Genome Association and Prediction Integrated Tool (GAPIT) in the R program 4.1.1.

Prior to the association analysis, principal component analysis (PCA) was performed to determine the optimal genetic clusters within GAPIT [62,63] to reduce false positives caused by population structures. However, PCA only accounts for fixed effects of genetic ancestry and does not account for relatedness between individuals. Therefore, a mixed-model approach [62], which used both fixed effects (candidate SNPs and fixed covariates) and random effects (the genotypic covariance matrix) involving kinship and cryptic relatedness, was used in the association analysis. The VanRaden method [64] was used to estimate the relationships among isolates by computing a kinship matrix.

To further confirm the clusters of genetically related isolates based on both the 209 SP-SNP markers used in the present study and also to compare the clusters with the 154 isolates using the 14 SSR markers in the previous study [17,36], a hierarchical cluster analysis was conducted using the dissimilarity values and the “ward.D2” method with the “hclust” function in the R stats 4.1.1 program [65].

Associations between SP-SNPs and the 18 virulence/avirulence traits of the *Pst* isolates were analyzed in the program GAPIT, using the commands “myGAPIT <- GAPIT (Y = myY, G = myG, PCA.total = “ ”, kinship.cluster = c(“ward.D2”), kinship.group = c(“Mean”), model = “MLM”, multiple_analysis = TRUE, NJtree.group = “ ”, and Geno.View.output = FALSE, file.output = T)”. The markers were identified as significant if the *p*-value was equal to or less than 0.01 and −log_10_ (*p*) equal to or greater than 2 after Bonferroni correction [66]. The Manhattan plots were drawn using the ‘CMplot’ package in the R program 4.1.1. The correlation relationships among the 18 virulence phenotypic traits were estimated using the ‘heatmap2’ package in the R program 4.1.1.

To determine how reliable the SP-SNP markers identified from the association analysis for detecting the avirulence/virulence phenotypes were, accuracies and sensitivities coupled with their 95% confidence intervals were calculated [67]. For this analysis, accuracy was measured as the proportion of correct predictions among all the predictions and sensitivity was measured as the proportion of predictions correctly identified by the test, these values showing how good the test was for detecting the avirulence or virulence phenotypes. If isolates had the avirulence genotype and the marker allele for avirulence or vice versa, the predictions of the phenotypes of the isolates by the marker were considered correct predictions (CPs). If the isolates had unmatched marker alleles and phenotypes, the marker predictions of phenotypes were considered incorrect predictions (ICPs). Accuracy = (Number of correct predictions)/(Number of all predictions). Sensitivity = (Number of correct predictions that are correctly identified by the test)/(Number of all correct predictions).

## 5. Conclusions

In this study, we identified 19 SP-SNP markers significantly associated with 12 avirulence genes: *AvYr1*, *AvYr6*, *AvYr7*, *AvYr9*, *AvYr10*, *AvYr24*, *AvYr27*, *AvYr32*, *AvYr43*, *AvYr44*, *AvYrSP*, and *AvYr76*. Some SP-SNP markers were significantly associated with two or more avirulence genes, suggesting that these avirulence loci are located in an avirulence gene cluster, consistent with our previous studies. The present study further confirmed that association analysis in combination with SP-SNP markers is powerful when it comes to identifying markers for avirulence genes. The virulence-related SP-SNP markers in the present study, as well as other SP-SNP markers identified in our previous study, should be useful in developing a set of virulence-related markers for monitoring changes in virulence and genotypes in *Pst* populations. The genomic resources developed in the present study can be used for further research on host–pathogen interactions as well as pathogen population dynamics.

## Figures and Tables

**Figure 1 ijms-23-04114-f001:**
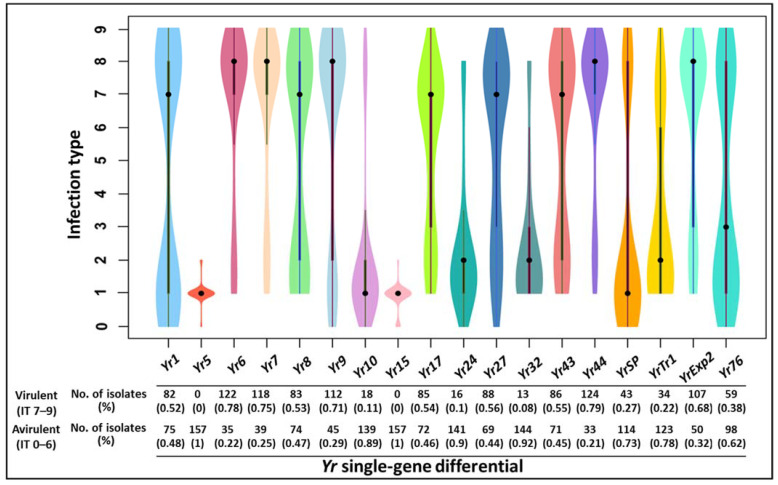
Violin plot showing distributions of infection type (IT) for 157 *Puccinia striiformis* f. sp. *tritici* isolates scored on 18 wheat lines with single *Yr* genes. Solid dots show medians. The numbers of isolates and the frequency values of the two phenotypic classes (avirulent and virulent) are given below the *Yr* genes.

**Figure 2 ijms-23-04114-f002:**
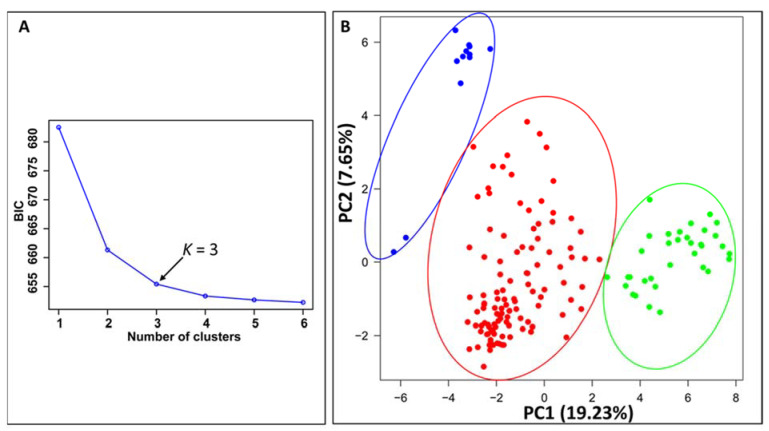
Principal component (PC) analysis of 157 *Puccinia striiformis* f. sp. *tritici* isolates using 209 SP-SNP markers. (**A**) The optimal *K* value (indicated by the black arrow) for determining the number of clusters based on the curve of Bayesian information criterion (BIC) values versus the number of clusters assessed with 209 SP-SNP markers. (**B**) Plot of the second principal component (PC2) against the first principal component (PC1) showing the three molecular groups. Each dot represents an isolate.

**Figure 3 ijms-23-04114-f003:**
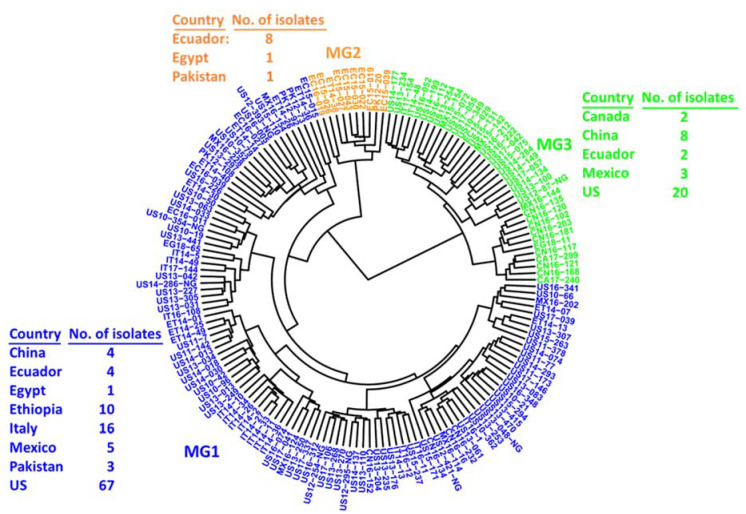
Dendrogram of *Puccinia striiformis* f. sp. *tritici* isolates from nine countries constructed based on dissimilarities assessed with 209 secreted protein gene-based SNP (SP-SNP) markers using hierarchical cluster analysis, showing three molecular groups (MGs) and the isolate numbers from different countries within each MG.

**Figure 4 ijms-23-04114-f004:**
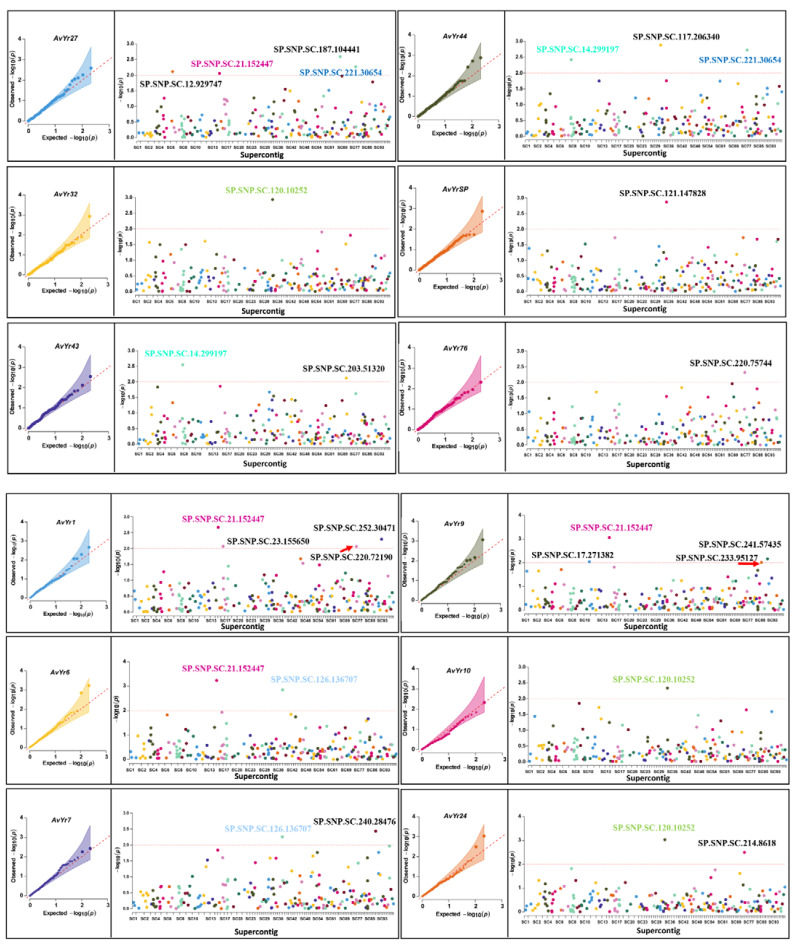
QQ plots and Manhattan plots of SP-SNP markers significantly associated with 12 avirulence (*Av*) genes. In the QQ plots, the *X*-axis represents the genomic position of SP-SNPs in the supercontigs of the PST-78 reference genome, and along the *Y*-axis are the −log_10_ transformed significance *p*-values. The red dashed lines represent the Bonferroni-corrected threshold −log_10_ (*p*) of 2.0. In the Manhattan plots, each dot represents a SP-SNP locus, and its genomic position is referred to the supercontig of the PST-78 reference genome. The SP-SNPs of same color are in the same supercontig.

**Figure 5 ijms-23-04114-f005:**
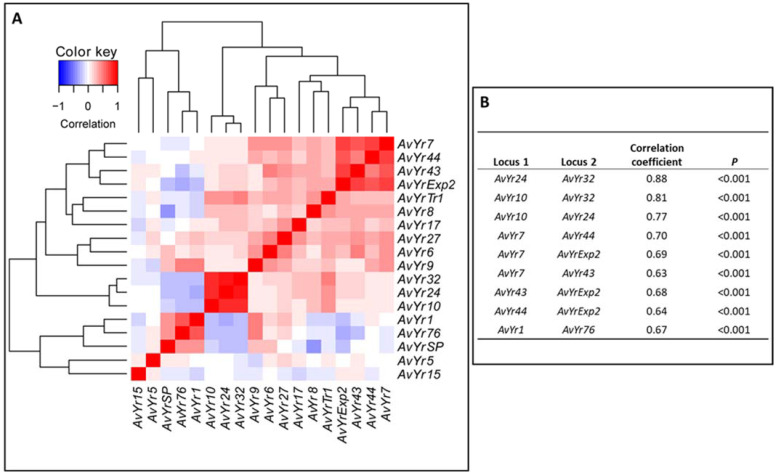
Correlation coefficients between 18 avirulence/virulence loci of *Puccinia striiformis* f. sp. *tritici* (**A**). The avirulence genes were symbolized as *AvYr1*, *AvYr5*, and so on, corresponding to their resistance *Yr* genes *Yr1*, *Yr5*, and so on. The correlations with coefficient values > 0.60 (*p* < 0.001) are listed in (**B**).

**Table 1 ijms-23-04114-t001:** SP-SNPs associated with avirulence genes in *Puccinia striiformis* f. sp. *tritici*.

Avirulence Gene ^a^	SNP ID	Supercontig ^b^	Position in Supercontig ^b^	*p*-Value	MAF ^c^	PVE ^d^	Allele ^e^	Protein ID ^b^
*AvYr1*	SP.SNP.SC.21.152447	21	152447	0.002144	0.38	0.15	C/G	PSTG_04155
	SP.SNP.SC.252.30471	252	30471	0.005105	0.11	0.14	G/T	PSTG_16039
	SP.SNP.SC.23.155650	23	15565	0.008536	0.42	0.13	T/G	PSTG_04466
	SP.SNP.SC.220.72190	220	7219	0.008644	0.45	0.13	A/G	PSTG_15512
*AvYr6*	SP.SNP.SC.21.152447	21	152447	0.000586	0.38	0.12	C/G	PSTG_04155
	SP.SNP.SC.126.136707	126	136707	0.001405	0.05	0.11	T/C	PSTG_12716
*AvYr7*	SP.SNP.SC.240.28476	240	28476	0.003675	0.49	0.06	G/A	PSTG_15854
	SP.SNP.SC.126.136707	126	136707	0.001405	0.05	0.11	T/C	PSTG_12716
*AvYr9*	SP.SNP.SC.21.152447	21	152447	0.000865	0.38	0.14	C/G	PSTG_04155
	SP.SNP.SC.241.57435	241	57435	0.006845	0.49	0.12	T/G	PSTG_15874
	SP.SNP.SC.17.271382	17	271382	0.009011	0.40	0.11	A/T	PSTG_03500
	SP.SNP.SC.233.95127	233	95127	0.009733	0.48	0.11	A/G	PSTG_15751
*AvYr10*	SP.SNP.SC.120.10252	120	10252	0.004658	0.25	0.07	C/T	PSTG_12413
*AvYr24*	SP.SNP.SC.120.10252	120	10252	0.000932	0.25	0.11	C/T	PSTG_12413
	SP.SNP.SC.214.8618	214	8618	0.003183	0.24	0.09	T/C	PSTG_15361
*AvYr27*	SP.SNP.SC.187.104441	187	104441	0.002582	0.10	0.10	A/C	PSTG_14812
	SP.SNP.SC.221.30654	221	30654	0.00544	0.37	0.09	A/T	PSTG_15517
	SP.SNP.SC.12.929747	12	929747	0.007666	0.33	0.09	G/A	PSTG_02640
	SP.SNP.SC.21.152447	21	152447	0.008759	0.38	0.09	C/G	PSTG_04155
*AvYr32*	SP.SNP.SC.120.10252	120	10252	0.001166	0.25	0.09	C/T	PSTG_12413
*AvYr43*	SP.SNP.SC.14.299197	14	299197	0.002819	0.08	0.10	C/T	PSTG_02897
	SP.SNP.SC.203.51320	203	5132	0.007604	0.31	0.08	T/G	PSTG_15141
*AvYr44*	SP.SNP.SC.117.206340	117	20634	0.001316	0.44	0.08	A/G	PSTG_12281
	SP.SNP.SC.221.30654	221	30654	0.00191	0.37	0.07	A/T	PSTG_15517
	SP.SNP.SC.14.299197	14	299197	0.003847	0.08	0.06	C/T	PSTG_02897
*AvYr76*	SP.SNP.SC.220.75744	220	75744	0.004842	0.46	0.17	A/T	PSTG_15513
*AvYrSP*	SP.SNP.SC.121.147828	121	147828	0.00136	0.44	0.21	C/A	PSTG_12496

^a^ Avirulence genes (*AvYr*) correspond to wheat *Yr* resistance genes. ^b^ Supercontig, position, and protein ID of the SP-SNP markers according to the reference genome PST-78 in the BROAD Institute Puccinia database (http://www.broadinstitute.org/, accessed on 24 November 2021). ^c^ MAF = minor allele frequency. ^d^ PVE = phenotypic variance explained by the significantly associated markers. ^e^ For each SP-SNP marker, the first allele was the major allele and the second allele was the minor allele.

**Table 2 ijms-23-04114-t002:** Numbers of the *Puccinia striiformis* f. sp. *tritici* isolates and races from nine countries in 2010–2018 used in this study.

Country	No. of Isolates	Year	Races ^a^
Canada	2	2017	PSTv-37, PSTv-14
China	12	2016	PSTv-225, PSTv-229, PSTv-230, PSTv-231, PSTv-250, PSTv-259, PSTv-267, PSTv-270, PSTv-274, PSTv-277, PSTv-278, PSTv-280
Ecuador	13	2015/2016	PSTv-20, PSTv-106, PSTv-221, PSTv-285, PSTv-286, PSTv-287, PSTv-289, PSTv-294, PSTv-298, PSTv-303, PSTv-305, PSTv-306, PSTv-327
Egypt	2	2018	PSTv-120, PSTv-15
Ethiopia	11	2014	PSTv-41, PSTv-47, PSTv-76, PSTv-105, PSTv-106, PSTv-107, PSTv-110, PSTv-116
Italy	18	2014/2016/2017	PSTv-121, PSTv-125, PSTv-127, PSTv-129, PSTv-130, PSTv-131, PSTv-132, PSTv-133, PSTv-134, PSTv-135, PSTv-136, PSTv-137, PSTv-192, PSTv-232, PSTv-295, PSTv-317, PSTv-320
Mexico	8	2015/2016	PSTv-53, PSTv-78, PSTv-109, PSTv-198, PSTv-252, PSTv-292, PSTv-296, PSTv-307
Pakistan	4	2012	PSTv-11, New, PSTv-37, New
USA	87	2010/2011/2012/2013/2014/2015/2016/2017	PSTv-1, PSTv-2, PSTv-3, PSTv-4, PSTv-6, PSTv-7, PSTv-8, PSTv-11, PSTv-14, PSTv-15, PSTv-16, PSTv-17, PSTv-18, PSTv-19, PSTv-20, PSTv-22, PSTv-23, PSTv-24, PSTv-25, PSTv-27, PSTv-28, PSTv-29, PSTv-31, PSTv-32, PSTv-33, PSTv-34, PSTv-35, PSTv-37, PSTv-39, PSTv-40, PSTv-41, PSTv-42, PSTv-43, PSTv-44, PSTv-45, PSTv-46, PSTv-47, PSTv-48, PSTv-52, PSTv-53, PSTv-64, PSTv-65, PSTv-67, PSTv-71, PSTv-72, PSTv-73, PSTv-74, PSTv-75, PSTv-76, PSTv-77, PSTv-78, PSTv-79, PSTv-101, PSTv-109, PSTv-120, PSTv-121, PSTv-122, PSTv-123, PSTv-144, PSTv-175, PSTv-198, PSTv-201, PSTv-214, PSTv-221, PSTv-239, PSTv-284, PSTv-293, PSTv-321, PSTv-322

^a^ The races were identified using the 18 *Yr* single-gene lines as differentials [13,14,15,39].

## Data Availability

The data presented in this study are available in the Appendix A.

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
