# Peer review of "Identification of Secreted Protein Gene-Based SNP Markers Associated with Virulence Phenotypes of Puccinia striiformis f. sp. tritici, the Wheat Stripe Rust Pathogen"

_ijms, 2022, doi:10.3390/ijms23084114_

Round 1
Reviewer 1 Report
The authors use molecular genotyping and principal component analysis with previously identified secreted protein gene SNPs in the important pathogenic blast fungus Puccinia striiformis f. sp. tritici (Pst) to identify associations with pathogenicity genes for virulence/avirulence on wheat. A useful feature of this work is the exploitation of relatively large numbers of SNPs and pathogen isolates. The experimental approaches are sound, the manuscript is well written, and the interpretations are reasonable. The findings are incremental, but are important steps toward the ultimate goal of identifying specific genes responsible for pathogenicity.
Line 41: “threat” should be “threaten”
Line 264: “chose” should be “chosen”
Author Response
Response: Thank you for the positive evaluation of this work.
- Line 40: “threat” should be “threaten”
Response: Corrected in line 40.
- Line 277: “chose” should be “chosen”
Response: Corrected in line 277.
Reviewer 2 Report
In the manuscript “ Identification of Secreted Protein Gene-based SNP Markers 2 Associated to Virulence Phenotypes of Puccinia striiformis f. sp. tritici, the Wheat Stripe Rust Pathogen” by Bai et al. the new genomic resources and possibilities for functional markers for Wheat Stripe Rust are addressed. This study provides more data on mechanisms of host-pathogen interactions, and tools for tagging specific virulence genes and race groups. Presented work is an element of the important larger study on wheat stripe rust secreted protein gene derived-SNPs published in various papers by the research group between 2016 and 2021.
In the present study, authors succeeded in the identification of SP-SNP markers for 12 out of the total 18 PSt avirulence genes corresponding to the 18 Yr single genes from different geographical sources. Markers for the remaining 6 genes still need to be discovered.
The manuscript represents novel data, well written.
I have some minor remarks:
Line 192: The accuracy of for the detections of avirulence/virulence were ranged from 50.97% in the test of the marker SP.SNP.SC.220.72190 with avirulence gene AvYr1 to 94.90% in the test of marker SP.SNP.SC.120.10252 with avirulence gene AvYr32 (Table S4).
Should be replaced by: The accuracy of for the detections of avirulence/virulence were ranged from 50.39% in the test of the marker SP.SNP.SC.187.104441 with avirulence gene AvYr27 to 94.90% in the test of marker SP.SNP.SC.120.10252 with avirulence gene AvYr32 (Table S4).
Line 201- The Figure 5 was divided into A and B subfigures, therefore the description must contain references to both parts.
Line 209 - “or only from the US”?
Line 240: Furter , replace by “further”
Line 264: was chose , change to: “was chosen”
Supplementary data:
Supplemetary Table S4. Summary of the phenotypes, genotypes, the interaction between phenotypes and genotype glass and the accuracy and sensitivity
Should be: Supplementary Table S4. Summary of the phenotypes, genotypes, the interaction between phenotypes and genotype class and the accuracy and sensitivity
Author Response
Response: Thank you for the positive evaluation of this work.
Minor remarks:
- Line 194: The accuracy for the detections of avirulence/virulence were ranged from 50.97% in the test of the marker SP.SNP.SC.220.72190 with avirulence gene AvYr1 to 94.90% in the test of marker SP.SNP.SC.120.10252 with avirulence gene AvYr32 (Table S4).
Should be replaced by: The accuracy of the detections of avirulence/virulence were ranged from 50.39% in the test of the marker SP.SNP.SC.187.104441 with avirulence gene AvYr27 to 94.90% in the test of marker SP.SNP.SC.120.10252 with avirulence gene AvYr32 (Table S4).
Response: Thanks for catching the error. We have corrected the sentence in line 194-196.
Line 201- Figure 5 was divided into A and B subfigures, therefore the description must contain references to both parts.
Response: Thanks for the suggestion. We have made the change in lines 213-216.
- Line 222 - “or only from the US”?
Response: We have added “and only from the U.S.” in line 222.
- Line 253: Furter , replace by “further”
Response: Corrected in line 253.
- Line 277: was chose, change to: “was chosen”
Response: Corrected in line 277.
Supplementary data:
- Supplemetary Table S4. Summary of the phenotypes, genotypes, the interaction between phenotypes and genotype glass and the accuracy and sensitivity
Should be: Supplementary Table S4. Summary of the phenotypes, genotypes, the interaction between phenotypes and genotype class and the accuracy and sensitivity
Response: Corrected the wrong word “glass” to “class” in line 503 and also in Table S4.
Reviewer 3 Report
Stripe rust caused by Puccinia striiformis f. sp. tritici (Pst) is a destructive wheat disease worldwide. The manuscript Identification of Secreted Protein Gene-based SNP Markers Associated to Virulence Phenotypes of Puccinia striiformis f. sp. tritici, the Wheat Stripe Rust Pathogen presents results of characterization of the US and international Pst isolates using the new SP-SNP markers and identification of SP-SNPs associated to avirulence genes.
The article clearly and in detail describes the methodology and results of the experiments. The results obtained using SP-SNP markers are tightly consistent with those obtained using microsatellite markers. Several SP-SNP markers were identified for significant associations with detected avirulence genes. This study in future can use for developing functional markers for tagging specific virulence genes and race groups that very actual for wheat breeding to this pathogen.
Author Response
Response: Thank you for the positive evaluation of this work.